# CFD Simulation of Dynamic Temperature Variations Induced by Tunnel Ventilation in a Broiler House

**DOI:** 10.3390/ani14203019

**Published:** 2024-10-18

**Authors:** Lak-yeong Choi, Kehinde Favour Daniel, Se-yeon Lee, Chae-rin Lee, Ji-yeon Park, Jinseon Park, Se-woon Hong

**Affiliations:** 1Department of Rural and Bio-Systems Engineering, Chonnam National University, Gwangju 61186, Republic of Korea; cly5002@gmail.com (L.-y.C.); kenniedee@jnu.ac.kr (K.F.D.); seyeonn@jnu.ac.kr (S.-y.L.); chaerinl@jnu.ac.kr (C.-r.L.); qkrwldus7749@jnu.ac.kr (J.-y.P.); 2Education and Research Unit for Climate-Smart Reclaimed-Tideland Agriculture (BK21 Four), Chonnam National University, Gwangju 61186, Republic of Korea; icarus381@jnu.ac.kr; 3AgriBio Institute of Climate Change Management, Chonnam National University, Gwangju 61186, Republic of Korea

**Keywords:** poultry heat production, thermal gradients, microclimate, inlet baffles, CFD validation

## Abstract

Maintaining the optimal temperature in broiler houses, where chickens are raised, is crucial for their health and productivity. However, efficiently managing indoor temperatures is challenging due to the large size of these facilities and varying weather conditions. This study developed and tested a computer model that predicts changes in air temperature inside a broiler house when ventilation systems are adjusted. The model accurately simulated how air movement affects indoor temperatures based on the number of fans operating or stopped. It also accounted for the heat produced by the chickens. The results also showed that adjusting the openings of air inlets can help reduce high temperatures but this can sometimes make the ventilation system less efficient or lead to an uneven temperature distribution. This study confirmed that this approach effectively improves ventilation strategies by comparing the model’s predictions with real-life data from broiler houses. This model can assist farmers and engineers in designing better ventilation systems, ensuring better conditions for chickens. Improved conditions will lead to more efficient poultry production, benefiting both the industry and consumers.

## 1. Introduction

Poultry production is an important sector within global food production that contributes substantially to food security. The rearing environment in broiler houses is crucial for the health, welfare, and productivity of the birds [1]. Key environmental parameters, such as air temperature and humidity, must be carefully controlled to provide optimal conditions for all birds [2]. However, maintaining these conditions consistently in commercial broiler houses is challenging due to varying external weather and the metabolic heat generated by the birds [3,4,5].

Ventilation systems are crucial for maintaining suitable indoor climates in broiler houses [6,7,8]. In South Korea, summer temperatures often exceed 30 °C while winter temperatures frequently drop below −10 °C. As a result, commercial broiler houses that operate year-round are designed with a strong emphasis on insulation. Broiler houses, in particular, often feature modern, enclosed facilities equipped with tunnel ventilation systems to alleviate thermal stress through the wind chill effect [9,10,11]. Tunnel ventilation works by drawing air through the length of the house via exhaust fans. In a conventional system, inlet baffles are typically operated uniformly, regardless of localised conditions, such as temperature gradients and uneven air distribution, which can negatively impact bird health and performance [12,13]. Thus, there is a need for more sophisticated ventilation methods that can adapt to the dynamic conditions within broiler houses.

One of the most effective ways of studying the ventilation and microenvironment of a broiler house is computational fluid dynamics (CFD) simulation [14,15,16,17]. It can help analyse and predict airflow behaviour and microclimate distribution within the indoor space. Furthermore, one significant advantage of CFD simulation is the ability to test various ventilation improvement methods in commercial broiler houses without physical trials, which can be costly and time-consuming [8,12,17,18].

When studying ventilation performance using CFD, simulating the heat and mass transport inside the broiler house is essential. It is crucial to model the sensible and latent heat the chickens release as they interact with the surrounding microclimate. Previous measurements have provided data curves describing the sensible and latent heat production of broilers aged two to seven weeks at 15.6 °C, 21.1 °C, and 26.7 °C [19]. The environmental chamber test presented the data curves for broiler chickens for the first four weeks and regression models as a function of chicken age or weight [20]. The data-based models were analysed in the form of a dynamic model using transfer functions [21,22]. Moreover, the heat transfer process in chickens (from the body core through the skin and coat to the surrounding environment) was represented as a thermal balance model following the physiological processes of the chickens [23,24,25]. This mechanistic model represented a significant advancement over empirical models.

The heat transfer models were integrated with the CFD models, which enable the dynamic calculation of heat generation based on environmental factors surrounding the chickens, such as air temperature and velocity, as determined by CFD simulation [25,26,27]. This coupling allowed for a more accurate prediction of the microclimate distribution within the broiler house and thus enabled the evaluation of the ventilation systems for improvement. Previous studies have demonstrated the utility of CFD in optimising ventilation designs [9,28] but there remains a gap in its application to dynamic control systems in broiler houses. While most studies have focused on evaluating the performance of ventilation systems under stable indoor and outdoor temperature conditions, in real-life commercial broiler houses, the operation of ventilation fans and the opening of inlets vary according to changes in indoor and outdoor temperatures. In particular, ventilation fans are often operated in an on–off cycle under low ventilation rates. There has been little CFD research to optimise these irregular ventilation operation methods or predict dynamic changes in the indoor microclimate under such operations. This study aims to develop a CFD model that is capable of predicting the temporal changes in indoor air temperature in response to variable ventilation operations in a commercial broiler house.

## 2. Materials and Methods

### 2.1. Experimental Farm

The experimental broiler house is located in Namwon, Jeollabuk-do, Republic of Korea. The building keeps 30,000 broilers with a floor area of 87 × 15 m^2^, a ridge height of 5 m, and an eave height of 3.5 m. It uses a tunnel ventilation system with 14 exhaust fans, each with a capacity of 37,000 m^3^ h^−1^. Cooling pads are installed on both sides of the building to alleviate thermal stress during summer (Figure 1). The rearing environment is automatically controlled by a microclimate control system (2nd Generation Smart Farm, Koko-farm system, Emotion Co., Ltd., Jeonju-si, Republic of Korea). The inlet baffles are installed along the length of both sides, with 29 openings on each side. While the tunnel fans are not operating, six circulation fans mounted in the ceiling pull air from below and circulate it throughout the indoor space.

The indoor space was divided into six zones, with two zones in the width direction and three in the length direction, to identify the spatial microclimate distribution (Figure 1). In addition, since tunnel ventilation systems tend to suffer from the longitudinal heterogeneity of the indoor microclimate, it is beneficial to measure the air temperature at multiple longitudinal points. The farthest zones from the tunnel fans were set as Zones 1 and 4, the middle zones were Zones 2 and 5, and the nearest zones to the tunnel fans were Zones 3 and 6. Air temperature was measured in the middle of each zone at a height of 0.6 m using thermometer sensors with a resolution of 0.1 °C and an accuracy of ±0.2 °C (LO-1000, Koko-farm system, Emotion Co., Ltd., Republic of Korea). The meteorological variables, including air temperature and humidity, were measured by a weather station installed on the experimental farm.

### 2.2. Development of the CFD Model

#### 2.2.1. Computational Domain

The full-scale, three-dimensional computational domain was created using ICEM-CFD (Ansys Inc., Canonsburg, PA, USA). All meshes were created as hexahedral cells. The interior of the broiler house was simplified by excluding complex components, such as feeders and pipes, and modelling only exterior walls. The tunnel fans were designed as squares with equivalent areas. The inlet baffles were modelled as volumetric spaces extruded 10 cm outward from the opening area instead of modelling the complex shapes of the baffles. This alternative method was implemented to avoid using dynamic meshes to change the angles of the baffles over time. Instead, it applied appropriate momentum loss within the volumetric spaces to simulate the varying open area of the inlets using user-defined functions (UDFs). In addition, the circulation fans were simplified into truncated cone shapes and the O-grid function of ICEM was used to generate hexahedral meshes for circular geometries (Figure 2). The circulation fans were designed to intake the air from the lower circular base and discharge it in 360° radial directions through the lateral surface.

#### 2.2.2. Modelling of Broiler Chickens

Geometrically modelling the shape of broiler chickens is significantly challenging due to the complexity of individual chickens and the variability in flock formations. As in previous studies [25], the volume from the floor of the broiler house (up to a height of 0.45 m) was defined as the bird zone to model the chickens. In the bird zone, both the sensible heat generated by the broilers and the air resistance caused by the chickens were included to simulate their impact on the microclimate within the broiler house.

The heat transfer process between chickens and their surrounding microclimate can be described using three consecutive layers, as shown in Figure 3 [23,25,29]. These layers are between the body core and the skin, between the skin and the hair coat, and between the hair coat and the surrounding air.

The heat transfer models within the three layers are thoroughly explained in Kim et al. [25] and Norton et al. [29]. In short, Equations (1)–(3) are the heat transfer in the first, second, and third layer, respectively:(1)$$Q_{t}=\frac{T_{b}-T_{s}}{r_{t}}\;,$$
(2)$$Q_{c}=\frac{T_{s}-T_{c}}{r_{c}}+E_{c},$$
(3)$$Q_{a}=\frac{T_{c}-T_{a}}{r_{a}},$$
where $Q_{t}$ denotes the heat loss from the body core to the skin (W m^−2^); $Q_{c}$ is the heat loss from the skin to the outer surface of the hair coat (W m^−2^); $Q_{a}$ signifies the convective heat loss from the hair coat to the air (W m^−2^); $T_{b}$, $T_{s}$, $T_{c}$, and $T_{a}$ are the mean temperatures of the body core, skin, outer surface of the hair coat, and air, respectively (°C); $r_{t}$, $r_{c}$, and $r_{a}$ denote the thermal resistance of the body tissue, hair coat, and boundary layer, respectively (m^2^ °C W^−1^); and $E_{c}$ is the latent heat loss from the skin (W m^−2^).

There are additional heat transfers other than the three processes: the bird’s respiration causes the latent heat loss directly from the body core to the air ($E_{r}$), which was estimated to be 5 W m^−2^ [24]. The conductive heat loss due to direct contact of the bird with the floor ($Q_{flr}$) and the radiative heat exchange between the bird and the surrounding objects ($Q_{r}$) are also significant and can be calculated as follows:(4)$$Q_{flr}=\frac{T_{s}-T_{flr}}{r_{t}+r_{flr}}\;,$$
(5)$$Q_{r}=\frac{T_{c}-T_{r}}{r_{r}},$$
where $T_{flr}$ is the temperature of the floor (°C); $T_{r}$ denotes the radiant temperature of the surroundings (°C); $r_{flr}$ is the thermal resistance of the floor (m^2^ °C W^−1^); and $r_{r}$ is the radiative resistance of the boundary layer (m^2^ °C W^−1^).

The above heat transfers can be calculated after the four temperatures are determined. While the body core temperature is constant at 41 °C and the air temperature can be derived from CFD calculations, the skin and hair coat temperatures are not explicitly known and can vary depending on the heat transfer conditions. These two unknown temperatures can be estimated through the following additional heat transfer relationships:(6)$$Q_{t}=R\left(Q_{c}+E_{r}\right)+\left(1-R\right)Q_{flr}\;,$$
(7)$$Q_{c}=Q_{a}+Q_{r},$$
where R is the proportion of the bird’s body exposed to the air.

All equations are interrelated and have nonlinear relationships; therefore, they can be solved using an iterative method. First, the initial values of $T_{s}$ and $T_{c}$ are set by dividing the temperature difference between $T_{b}$ and $T_{a}$ into three equal parts. Then, by substituting Equations (3)–(5) into Equations (6) and (7), $Q_{c}$ and $Q_{t}$ are calculated. Next, $T_{s}$ and $T_{c}$ are updated using Equations (1) and (2). With the updated $T_{s}$ and $T_{c}$, $Q_{c}$ and $Q_{t}$ are recalculated and this process is repeated until $T_{s}$ and $T_{c}$ converge. Then, the sensible heat generated by the chickens and transferred to the air, $Q_{a}$, is calculated using the final converged $T_{c}$ and Equation (3).

The rate of sensible heat generation from the birds to the air in the broiler house can be derived and used for the source term as follows:(8)$$S_{h}=\frac{A_{c}n_{b}Q_{a}}{V_{z}}\;$$

Here, $S_{h}$ is the source term for sensible heat generation (W m^−3^); $A_{c}$ is the surface area of the bird (m^2^); $n_{b}$ is the number of birds in the building; and $V_{z}$ is the volume of the bird zone in the building (m^3^).

The air resistance of the chickens can be estimated by the drag force acting on the birds due to the airflow, which is the identical force that the bird exerts on the airflow but in the opposite direction. The momentum loss in the bird zone due to the air resistance is derived as:(9)$$S_{u}=-\frac{n_{b}A_{p}C_{d}}{V_{z}}\frac{1}{2}\rho v^{2}\;\;$$

Here, $S_{u}$ is the source term for momentum loss in the bird zone (kg m^−2^ s^−2^); $A_{p}$ is the projected area of the chicken body (m^2^); $C_{d}$ is the drag coefficient of the chicken; ρ is the air density (kg m^−3^); and v is the air velocity (m s^−1^). Additional equations and parameters for broiler chickens are presented in Kim et al. [25] and Norton et al. [29].

#### 2.2.3. Boundary Conditions and Numerical Models

The exhaust fans were set as velocity-inlet with a negative wind speed to induce ventilation flow from the indoor space to the outside. The ventilation rate through the fans decreases as the pressure difference (before and after the fan) increases. The performance curve of the ventilation fans was provided by the manufacturer and incorporated into the CFD model using UDFs.
(10)$$V_{T}=1000\times \left(3.402\times \sqrt{38.650-0.588\times P_{V}}+15.850\right)/A_{T}$$

Here, $V_{T}$ is the wind speed at the ventilation fan (m s^−1^); $P_{V}$ denotes the pressure difference before and after the fan (N m^−2^); and $A_{T}$ is the area of the ventilation fan (m^2^).

The circulation fans operate by pulling the air from the lower zone of the building upwards and dispersing it laterally near the ceiling. The bottom surface of the circulation fans was set as a velocity-inlet with a negative wind speed and the side surfaces were set as velocity-inlets with a positive wind speed. The inflow velocity to the circulation fans was measured as 4.4 m s^−1^ and assumed constant because the pressure difference across the circulation fans could be negligible during broiler rearing.

The inlet baffles on both side walls were modelled as volumetric spaces. As the inlet baffle opens, outdoor air enters the building through the opening. The outer boundary of the inlet volume space was set as the pressure-inlet, which allows the outdoor air to be driven into the building by the pressure difference. Simultaneously, the closed area had to block the penetration of outdoor air, which was modelled by applying momentum loss to the closed area using the UDFs (Figure 4).
(11)$$S_{u,\;inlet}=-\frac{\rho v}{∆t}\;\;$$

Here, $S_{u,\;inlet}\;$is the source term for momentum loss in the inlet volume space (kg m^−2^ s^−2^) and ∆t denotes the length of the time step (s).

The cooling pads were designed as porous walls with a porous jump function and incorporated using UDFs to account for air resistance and temperature decrease. In addition, the inertial resistance for the cooling pad performance parameters was determined to be 23.3 m^−1^ through trial and error by comparing with the measurement results described in Section 2.4. The cooling efficiency was assumed to be 0.7 as cooling pads typically have an efficiency of around 70%. All exterior walls were set as no-slip walls and their heat transmission coefficient was set as 0.247 W m^−2^ K^−1^. The heat transmission coefficients of the walls were measured at the experimental farms three times using a gSKIN U-Value kit (GreenTeg, Rümlang, Switzerland) and calculated following the ISO-9869 [30].

Furthermore, air flows were solved using the Reynolds-averaged Navier-Stokes (RANS) equations with a pressure-based solver [6,25]. The buoyancy effects on airflow due to air temperature differences were simulated using the Boussinesq model. The turbulence quantities were modelled using the RNG k-ε model, which showed reliable accuracy over the wide range of air velocity in poultry buildings [31]. The standard wall functions were used for the near-wall treatment. In addition, the SIMPLE algorithm was used for pressure-velocity coupling. The second-order scheme discretised the pressure, momentum, turbulent quantities, and energy. The simulation runs transiently with a time step of 1 s. The convergence criteria for the energy and other variables were set as 1.0 × 10^−6^ and 1.0 × 10^−3^, respectively.

### 2.3. Grid Independence of the CFD Model

The independence of the grid was examined by comparing six different grid sizes to find the optimal grid size that balanced reasonable computational resources and acceptable accuracy. The number of grids was increased incrementally from 99,004 cells (Case 1) to 16,522,880 cells (Case 6). Steady-state CFD simulations were performed and the results of Cases 1 to 5 were compared to those of Case 6 by calculating the R^2^ values. Five tunnel fans were operated for convenience and the inlet baffles were fully open during the simulations.

From the converged CFD results, velocity and pressure data were obtained at the height of one meter from the floor at the centre of each of the six zones. The comparison showed high R^2^ values (0.94 or more) in most results (Table 1). However, Case 4 was selected as the appropriate grid size for transient simulations (Figure 5) because it had a sufficiently high number of cells and showed a high R^2^ value (0.99) for pressure, which indicates excellent agreement with Case 6. The minimum orthogonal quality should typically be 0.1 or higher and, in Case 4, it was 0.188, indicating a satisfactory quality level.

### 2.4. Measurements and CFD Model Validation

The validation of the CFD model was conducted in two steps. The first step verified the airflow and air velocity due to ventilation in an empty broiler house without chickens. While operating five or fourteen tunnel fans, the air velocity, in the longitudinal direction to align with the tunnel ventilation flow, was measured at the centre of each of the six zones at a height of 0.6 m from the floor. Simultaneously, the airflow entering the inlet baffles along the side walls and the cooling pads was measured. Four inlet baffles, which were evenly spaced along the length of the house, were selected and labelled as A1, A2, A3, and A4, starting from the inlet that is furthest from the tunnel fans. At each inlet, six anemometers were arranged in a grid pattern to measure the average air velocity of the incoming air evenly. The airflow rate was then calculated by multiplying the measured average air velocity by the inlet area. Hot-wire anemometers with a resolution of 0.01 m s^−1^ and an accuracy of 0.1 m s^−1^ (Testo 405i, Testo Instruments, Germany) were used in all measurements.

Similarly, three evenly spaced points were selected within the cooling pad and labelled C1, C2, and C3. The airflow entering through the cooling pad was observed only when 14 tunnel fans were in operation and was measured using a volume flow hood (Testo 420, Testo Instruments, Baden-Württemberg, Germany). All measurements were conducted on one side of the broiler house where Zones 1, 2, and 4 were located. The results from the field measurements were compared to the results of the CFD simulation under the same conditions to validate the CFD model.

The second step involved verifying the heat generation from the chickens and the resulting changes in indoor temperature due to ventilation in a broiler house with chickens present. The experimental data, which were collected on 24 March 2023 (between 18:00 and 19:00) were used. At the time of the experiment, the broilers were 21 days old, with a total of 30,000 birds being reared. During the experimental period, the operation status of the broiler house, including the tunnel fans, opening of the inlet baffles, indoor temperatures in six zones, and outdoor temperature, was continuously measured and recorded in real-time. Specifically, the operation data of the tunnel fans and inlet baffles were stored in JSON (JavaScript Object Notation) format, which was then converted into time-series data at one-second intervals (using MATLAB R2021a) for further analysis. In the CFD simulation, the recorded outdoor temperature, tunnel fan operation, and inlet baffle positions were used as inputs and the indoor temperatures in the six zones were simulated for comparison.

### 2.5. Variable Inlets for Controlling Thermal Gradients

Temperature gradients along the length of the building are inevitable in the tunnel ventilation system. This study explored the possibility of controlling these longitudinal temperature gradients by selectively adjusting the inlet baffles arranged along the length of the broiler house. The six zones were grouped into three sections: Zones 1 and 4, 2 and 5, and 3 and 6. In the current system, all inlet baffles are operated uniformly, but this study explored the impact of varying the degree to which the inlet baffles open along the length of the building.

In Case 1, the inlet baffles on the farthest section from the tunnel fans (Zones 1 and 4) were opened less while the baffles closest to the tunnel fans (Zones 3 and 6) were opened more. Conversely, in Case 2, the inlet baffles near the tunnel fans (Zones 3 and 6) were opened less and those farthest from the fans (Zones 1 and 4) were opened more. Furthermore, the inlet baffles in the middle section (Zones 2 and 5) were opened as they are in the current system. The “less open” baffles were set to 50% of the opening area of the middle section and the “more open” baffles were set to 150% of the opening area of the middle section. This ensured that the total open area of the inlets remained constant in the current system.

Following the same procedure as the validation simulation, the inlet baffle and tunnel fan operations were measured for one hour. During these simulations, Case 1 and Case 2 variable inlet baffle configurations were applied. The resulting temperature changes in each zone were then compared and analysed to evaluate the effectiveness of these configurations in controlling thermal gradients.

## 3. Results

### 3.1. CFD Model Validation for Air Velocity

The indoor air velocity distributions at a height of one meter from the floor (predicted by the CFD simulations) are shown in Figure 6. Overall, the air velocity increased as the distance to the tunnel fans decreased, which suggests a trend of faster airflow closer to the fans. The comparison between the measured and simulated air velocities in the six zones is shown in Table 2. The error ranged from −0.22 to 0.05 m s^−1^ when five tunnel fans were operating and from −0.04 to 0.32 m s^−1^ when fourteen tunnel fans were operating. On average, the predicted indoor air velocities from the CFD simulations demonstrated reasonable accuracy, with errors ranging from approximately −0.2 to 0.3 m s^−1^. Particularly, considering the anemometer’s accuracy of 0.1 m s^−1^, these results are within an acceptable range.

Table 3 compares the airflow rates entering the broiler house through the inlet baffles and cooling pads driven by the tunnel fans. There was a slight discrepancy between the measured and CFD-predicted airflow rates at the inlet baffles located at A1, A2, A3, and A4. When five tunnel fans were operating, the measurements showed higher airflow rates at A2 and A3 while the CFD results revealed more uniform airflow across all four locations. However, when 14 tunnel fans were operating, both the measured and CFD results showed the same trend of increasing airflow rates as the distance to the tunnel fans decreased.

After interpolating the measured results across all 58 inlets to calculate the total airflow rate and comparing it with the CFD predicted total airflow rate, the overall accuracy was approximately 97.4% and 108.1% for scenarios with five and fourteen tunnel fans operating, respectively. This marks a good agreement. Furthermore, when the cooling pads served as additional inlets for 14 tunnel fans operating, the total airflow rates through the cooling pads calculated from the measurements at C1, C2, and C3 showed good consistency with the CFD results with an accuracy of approximately 106.0%.

### 3.2. CFD Model Validation for the Air Temperature Variation

During the experiment, from 18:00 to 19:00, the outdoor temperature gradually decreased from 14.8 °C to 12.7 °C while the relative humidity remained nearly steady (at around 60%). Furthermore, the target temperature for broilers, which the farm owner set, was 26.5 °C. In order to maintain this temperature, the tunnel fans were controlled using a frequent on–off cycle. The number of operating tunnel fans and the degree of inlet baffle openings were recorded—see Figure 7. This intermittent operation of the tunnel fans can potentially be more advantageous for CFD model validation than a scenario where the fans operate continuously. When the tunnel fans were in the ‘on’ state, the validation of the ventilation effect was allowed due to forced ventilation by the fans. Conversely, when the fans were ‘off’, the indoor temperature increase due to the sensible heat generation by the chickens was validated.

Figure 8 shows the temperature variations across the six zones. According to the measured results, the temperatures in Zones 3 and 6, which are closer to the tunnel fans, were approximately 1.5 °C higher than in Zone 1 and Zone 4, which are farther from the tunnel fans. This reflects the typical thermal gradients observed in tunnel ventilation systems. As the warm air within the broiler house moves along the length of the building and is expelled through the tunnel fans, the highest temperatures are recorded near the fans [32,33]. When the inlet baffles opened and the tunnel fans were run, the indoor temperatures in all six zones decreased quickly. The degree of temperature drop was correlated with the duration of the tunnel fan operation. As shown in Figure 4, during the periods when the tunnel fans were operated for a relatively longer time (specifically at 18:04, 18:15, 18:27, 18:42, and 18:58), the indoor temperature dropped by up to approximately 3 °C.

The indoor temperature across the six zones computed by the CFD simulations with the identical operational changes of the tunnel fans and inlet baffles, shown in Figure 3, are presented in Figure 8. Overall, the temperature variation trends in the CFD results were nearly identical to those observed in the measurements. More specifically, the rate and magnitude of temperature changes during the sharp drops (when the tunnel fans were operating) and the gradual increases (when the fans were off) matched the measured data well. In addition, the root-mean-square error (RMSE) between the measured and CFD-computed average temperatures across the six zones was calculated as 0.50 °C. Considering the thermometer’s accuracy of ±0.2 °C, these results are within an acceptable range. This also suggests that the CFD model is sufficiently accurate to simulate indoor temperature response to tunnel fan operation.

However, the temperature differences between the six zones were relatively small in the CFD results compared to the measurements. There are two possible reasons for this discrepancy: 

First, while the tunnel ventilation system influences thermal gradients in the broiler house, they are also affected by unintended infiltration. In the experimental broiler house, the infiltration was particularly significant near the far end of the house, opposite the tunnel fans, where a large door was located to allow trucks to enter during bird loading. Even when the door was closed, significant air infiltration occurred through the gaps, which led to lower temperatures around the far-end wall. Since the CFD model did not account for infiltration, the thermal gradients caused by this effect were not reflected. 

Second, the initial conditions in the CFD model did not adequately map the real-life indoor temperature distribution. The CFD model used a uniform initial temperature across all zones while the measured data already showed temperature differences between the zones at 18:00. As a result, the simulated temperature gradient along the length direction was not significant until 18:27. Conversely, in the early periods of the simulation, the temperatures in Zones 1 and 4 were even higher than in the other zones. This occurred because these zones were located farther from the tunnel fans, which resulted in reduced ventilation effects and less influx of outdoor cold air. In addition, they were not influenced by infiltration in the CFD simulations.

Nevertheless, as the CFD simulation progressed, thermal gradients between 18:27 and 18:42 were well simulated, with Zones 3 and 6 showing clearly higher temperatures than Zones 1 and 4. After 18:42, the simulated thermal gradients became very weak again. However, it should be noted that thermal gradients along the length of the house, with temperatures rising toward the tunnel fans, did not consistently appear in the measured data either. Among the measured temperatures between 18:10 and 18:27, the highest temperature was recorded in Zone 1, which was farthest from the tunnel fans, while the lowest temperature was observed in Zone 6, which was closest to the tunnel fans. It was assumed that thermal gradients became more noticeable when the tunnel fans operated for a more extended period, as observed around 18:27, which allowed sufficient airflow to develop within the building.

### 3.3. Variable Inlets to Control Thermal Gradients

The variable opening of the inlet baffles altered the airflow paths of the incoming air. Figure 9 shows the indoor airflow paths from the inlet baffles in Cases 1 and 2. In Case 1, the jets significantly drew the air entering through all inlets into the interior, which descends into the bird zone. However, in Zones 1 and 4, which were farther from the tunnel fans and had less-open inlet baffles, the jets formed were slower compared to other zones. In Case 2, the inlet baffles in Zones 1 and 4, which were farthest from the tunnel fans, were opened to 150%, which resulted in the formation of relatively strong jets. On the other hand, in Zones 3 and 6, where the inlet baffles were less open, the air quickly descended into the bird zone, which sharply decreased air velocity.

Figure 10 shows the changes in indoor temperature before and after the tunnel fans were operated. The figures show well the increase in indoor temperature due to the birds’ heat generation when the tunnel fans were off (at 18:25 and 18:26), as well as the decrease in temperature due to ventilation during 18:27 and 18:28. In Case 1, as the inlet baffles were opened less in zones farther away from the tunnel fans, the ventilation effect was reduced, which resulted in higher indoor temperatures in those areas. When ventilation occurred, the indoor temperature dropped significantly in the central sections as well as in the zones near the tunnel fans (Zones 2, 5, 3, and 6). However, in Zones 1 and 4, which were farther away from the fans, the indoor temperature did not decrease as much, even during ventilation. This reflects the typical temperature gradient in tunnel ventilation systems, where temperatures tend to be higher closer to the tunnel fans. Case 1 showed benefits in cooling near the tunnel fans, which suggests that this inlet configuration could help address the challenge of temperature gradients. Nevertheless, the problem of insufficient ventilation in the sections farthest away from the tunnel fans arose, although this could potentially be mitigated by infiltration under real-life farming conditions.

In Case 2, where the inlet baffles were opened more in the zones farther away from the tunnel fans, the distant sections showed a stronger ventilation effect and the lowest temperatures while the zones closer to the fans exhibited the highest temperatures. However, during the ventilation period at 18:28, the ventilation effect was more uniform and decreased the indoor temperature across all zones along the length of the building.

The results of simulating the indoor temperature changes over one hour with on–off ventilation, based on the variable inlets in Case 1 and Case 2, are shown in Figure 11. The overall temperature distribution was similar to the observation from Figure 6. However, in Case 1, the temperatures in Zones 1 and 4 remained consistently higher, with minimal ventilation effect even during the ventilation periods. In Case 2, although the temperature gradient was not resolved and became greater, the temperatures dropped below 24 °C during the heavy ventilation periods at 18:04, 18:15, and 18:27, which indicates overall higher ventilation performance. The variations in average temperatures across the six zones are shown in Figure 12. Regardless of the spatial temperature differences between each zone, the overall average temperature in the bird zone was lowest in Case 2 and highest in Case 1. This suggests that while the inlet configuration in Case 1 effectively alleviated high temperatures near the tunnel fans, it reduced the overall ventilation effect. Conversely, the inlet configuration in Case 2 enhanced the overall ventilation effect, which not only led to a significant reduction in the average temperature across the entire bird zone but also risked intensifying the temperature gradient.

Even though the total volume of incoming air remained consistent across variable inlet configurations, the key differences lay in which section of the building received the most airflow. In Case 2, more air was introduced into the sections farthest away from the tunnel fans, which enabled the air to sweep through the entire building before being exhausted, thereby maximising the ventilation effect. In Case 1, however, more air entered the sections closer to the tunnel fans, which resulted in effective ventilation only in those areas near the fans before the air was exhausted. Consequently, the area within the building affected by the incoming air was the key factor, which led to the observed variations in ventilation effectiveness and temperature distribution.

## 4. Conclusions

This study successfully developed and validated a CFD model capable of predicting temporal changes in indoor air temperature in response to variable ventilation operations in a commercial broiler house. The model accurately simulated air velocity and airflow distribution with different numbers of operating tunnel fans, which demonstrated good agreement with measured data. The error in air velocity ranged from −0.22 to 0.32 m s^−1^, and the total airflow rates through the inlet baffles and cooling pads matched the measured values with an accuracy of up to 108.1%. This number confirms the model’s capability to predict ventilation performance. The CFD model also correctly predicted the temperature dynamics within the broiler house, including the effect of chicken heat production and ventilation patterns on the indoor microclimate. The simulation successfully predicted temperature gradients and their fluctuations during ventilation cycles. In the validation process, which involved comparing airflow velocities and rates in an empty house as well as thermal processes in a house with chickens, the CFD model’s error was within an acceptable range when compared to the measurement device accuracy, confirming the model’s accuracy and reliability. This study also explored different configurations of variable inlet baffles to control temperature gradients. Case 1, with reduced airflow in zones farther away from the fans, mitigated high temperatures near the fans but reduced overall ventilation efficiency. In Case 2, increased airflow in these zones improved ventilation effectiveness but increased temperature gradients. This demonstrates the trade-offs involved in designing ventilation strategies.

In conclusion, the developed CFD model proves to be a valuable tool for simulating ventilation and temperature changes in broiler houses. By analysing various ventilation configurations in a virtual environment, this model can significantly enhance the microenvironment of the bird zones and, thereby, improve poultry production outcomes.

## Figures and Tables

**Figure 1 animals-14-03019-f001:**
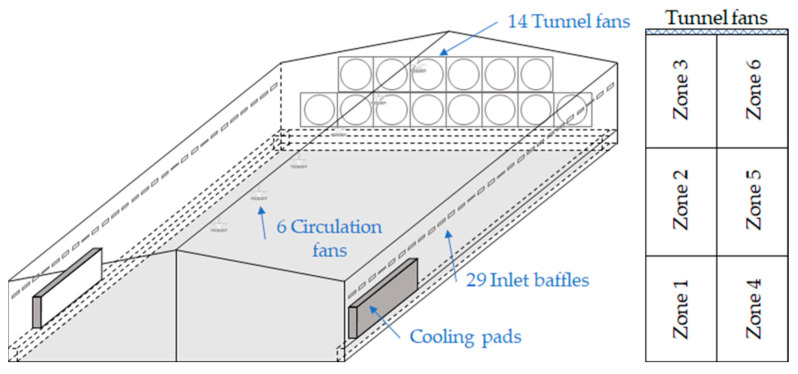
Schematic of the experimental broiler house and structure.

**Figure 2 animals-14-03019-f002:**
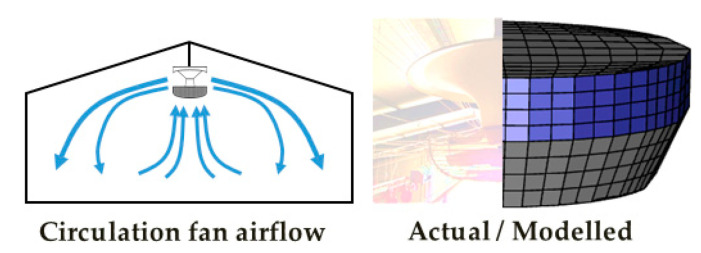
Airflow generated by circulation fans and mesh creation using an O-grid for these fans. The meshes in blue represent the lateral surface for air discharge.

**Figure 3 animals-14-03019-f003:**
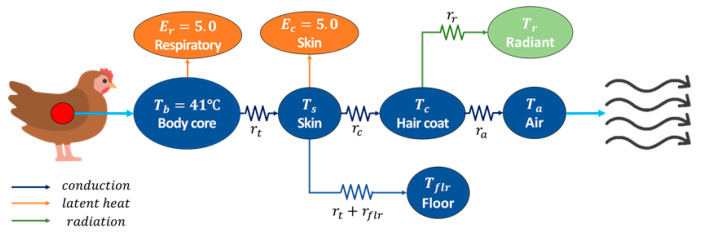
Schematic illustrating the heat transfer between broiler chickens and the surrounding air.

**Figure 4 animals-14-03019-f004:**
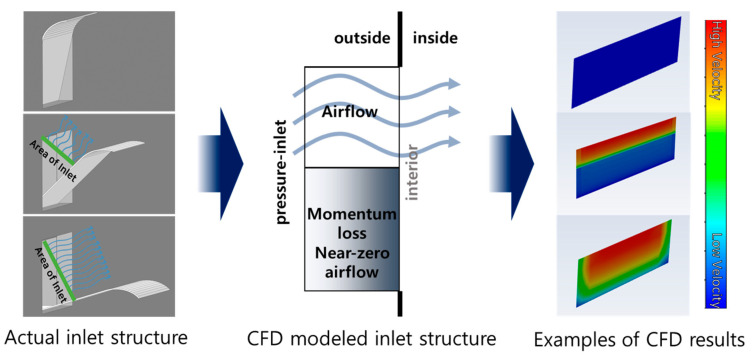
Modelling of the inlet baffles using air resistance.

**Figure 5 animals-14-03019-f005:**
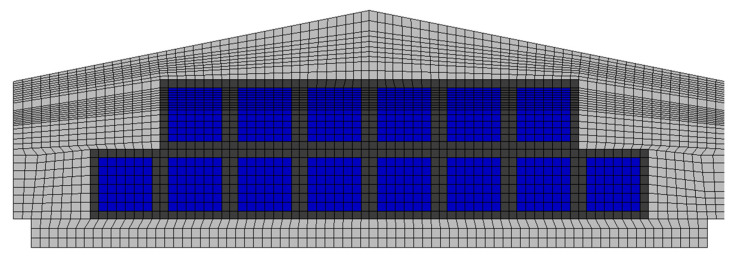
Mesh of Case 4, which satisfies the criteria for grid independence. The blue squares correspond to the exhaust fans, while the dark regions represent the fan frames.

**Figure 6 animals-14-03019-f006:**
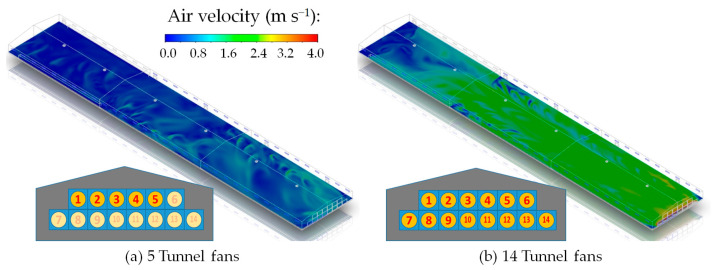
Indoor-air velocity distribution one meter above the floor with five or fourteen tunnel fans operating. The tunnel fans are numbered from 1 to 14, and the tunnel fans not in operation are shown faintly.

**Figure 7 animals-14-03019-f007:**
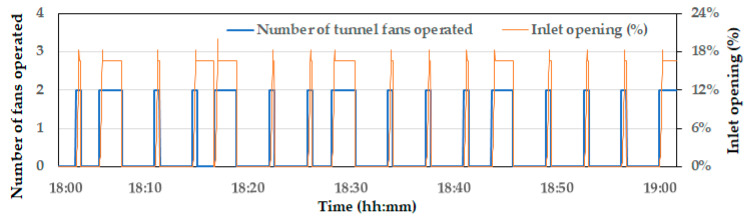
Operational status of the tunnel fans and inlet baffles during the experiment.

**Figure 8 animals-14-03019-f008:**
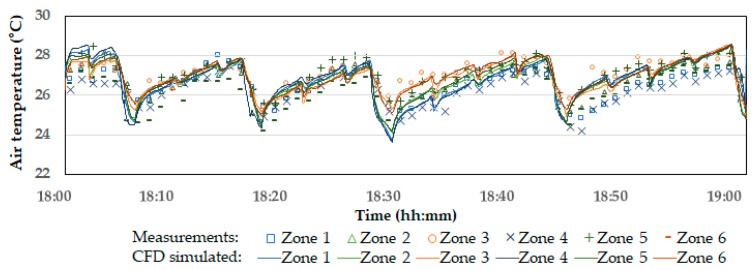
Comparison of measured indoor temperatures and CFD model predictions for the six different zones.

**Figure 9 animals-14-03019-f009:**
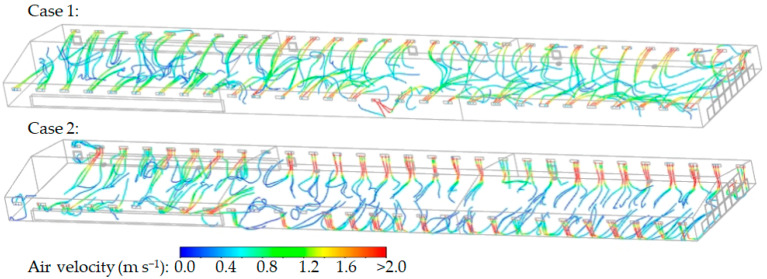
Pathlines illustrating the indoor airflow from inlet baffles (results at 18:28).

**Figure 10 animals-14-03019-f010:**
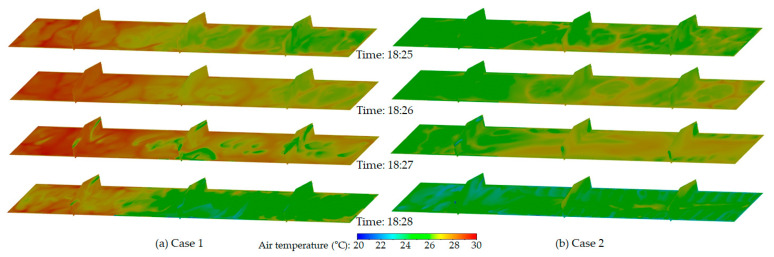
Indoor temperature distribution from 18:25 to 18:28 for two variable inlet configurations: (**a**) Case 1, featuring more open inlet baffles in the zones near the tunnel fans, and (**b**) Case 2, with more open inlet baffles in the zones farther from the tunnel fans. Contours are displayed on a horizontal plane 0.6 m above the floor and three vertical planes along the length of the building. Tunnel ventilation was active during 18:27 and 18:28.

**Figure 11 animals-14-03019-f011:**
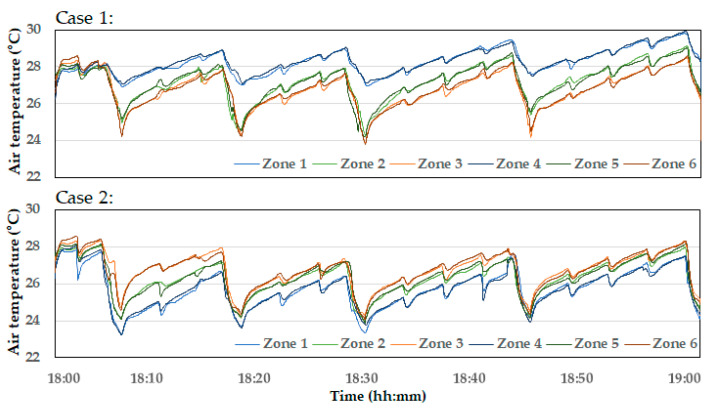
Indoor temperature changes in six zones due to variable inlets in Case 1 and Case 2.

**Figure 12 animals-14-03019-f012:**
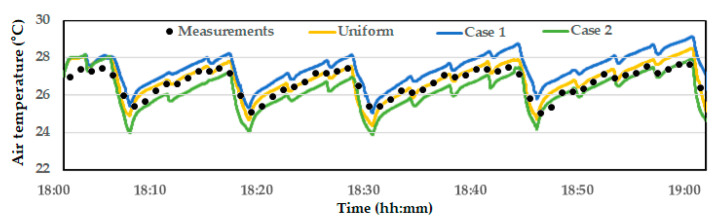
Comparison of average computed indoor temperature across six zones for different variable inlet configurations.

**Table 1 animals-14-03019-t001:** Comparison of the grid-independence test results for different grid sizes. R^2^ values represent comparisons with the smallest grid size (Case 6).

Grid Case	Size (m)		Number of Cells	Velocity R^2^	Pressure R^2^
	Minimum	Maximum			
1	0.234	1.270	99,004	0.94	0.97
2	0.224	0.641	392,976	0.94	0.98
3	0.224	0.499	997,848	0.95	0.97
4	0.184	0.392	2,065,360	0.96	0.99
5	0.112	0.297	7,620,576	0.97	1.00
6	0.092	0.250	16,522,880	1.00	1.00

**Table 2 animals-14-03019-t002:** Comparison of air velocity at the centre of six zones, with five and fourteen tunnel fans operating (unit: m s^−1^).

Locations	5 Tunnel Fans			14 Tunnel Fans		
	Measured	CFD Result	Error	Measured	CFD Result	Error
Zone 1	0.25	0.29	−0.04	1.38	1.06	0.32
Zone 2	0.5	0.51	−0.01	1.98	1.77	0.21
Zone 3	0.68	0.9	−0.22	2	2.04	−0.04
Zone 4	0.33	0.28	0.05	1.33	1.01	0.32
Zone 5	0.57	0.62	−0.05	1.82	1.77	0.04
Zone 6	0.84	0.9	−0.06	2.11	2	0.11

**Table 3 animals-14-03019-t003:** Comparison of incoming airflow through inlet baffles and cooling pads with five or fourteen tunnel fans operating (unit: m^3^ h^−1^ for A1, A2, A3, A4, inlet total, and cooling pad total; m^3^ h^−1^ m^−1^ for C1, C2, and C3).

Locations	5 Tunnel Fans		14 Tunnel Fans	
	Measured	CFD Result	Measured	CFD Result
A1	1763	2558	2711	2029
A2	3696	2704	3565	3453
A3	3750	2880	3361	3765
A4	2664	2855	3666	3961
Inlet total	175,908	171,397	185,912	200,923
C1	-	-	4995	5143
C2	-	-	5995	4948
C3	-	-	6230	6830
Cooling pad total	-	-	263,134	278,990

## Data Availability

The data presented in this study are available on request from the corresponding author. The data are not publicly available due to privacy concerns.

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
