# Peer review of "CFD Simulation of Dynamic Temperature Variations Induced by Tunnel Ventilation in a Broiler House"

_animals, 2024, doi:10.3390/ani14203019_

Round 1
Reviewer 1 Report
Comments and Suggestions for Authors
The grid size is too large. In future studies, I recommend making the grid denser. The mesh quality in Orthogonal Quality is not specified. This parameter is very important for CFD modeling. In future studies, I recommend using the Species Transport model (ANSYS Fluent). It will allow to assess the air quality by dividing it into chemical elements. To model infiltration, you can use the Porous Zone model (ANSYS Fluent). But this is not mandatory. The article does not require global changes; the research results are important and necessary.
Author Response
Comments 1: The grid size is too large. In future studies, I recommend making the grid denser. The mesh quality in Orthogonal Quality is not specified. This parameter is very important for CFD modeling.
Response 1: Thank you for pointing this out. Grid size is a crucial factor in CFD analysis. In this study, a simplified mesh was necessary due to the long-term transient analysis. Consequently, a grid independence test was performed, and case 4 was determined to have the most appropriate mesh size. In future research, we may consider trying a denser mesh based on the reviewer's suggestion. The minimum orthogonal quality was 0.188, and since this value generally needs to be 0.1 or higher, the mesh quality was deemed sufficient. This information has been added to the manuscript.
“The minimum orthogonal quality should typically be 0.1 or higher, and in case 4, it was 0.188, indicating a satisfactory quality level.”
Comments 2: In future studies, I recommend using the Species Transport model (ANSYS Fluent). It will allow to assess the air quality by dividing it into chemical elements.
Response 2: We agree with this comment. Air quality is an important issue, and in future research, when considering the distribution of factors such as ammonia or humidity, we plan to use the Species Transport model as suggested by the reviewer. However, since this study focused on temperature changes, it was not necessary to use the Species Transport model.
Comments 3: To model infiltration, you can use the Porous Zone model (ANSYS Fluent). But this is not mandatory.
Response 3: We agree with this comment. In previous research, we have used the porous zone model to simulate infiltration. However, I recall that determining the appropriate porosity values to match the actual infiltration rates posed significant challenges. If the impact of infiltration is substantial and we aim to incorporate it into the CFD model, additional experiments to determine the correct porosity values would be necessary. While not mandatory, as the reviewer mentioned, we believe this is something worth considering in future studies.
Reviewer 2 Report
Comments and Suggestions for Authors
Detailed comment
Line 48: Introduction: The objective, previous research and state of the art described in the introduction are well presented and in a very concise way.
Line 102: Experimental farm: The description of the experimental farm is precise and well-illustrated while concise. However, it lacks the description and precision of the microclimate measurements and particularly the types of devices used for air speed determination. Particularly, are they 1, 2, 3 dimensional or omnidirectional (only the modulus of air speed is measured?). Give also the characteristics and the precisions of the air temperature and humidity measurements and use these data in the discussion of the model errors range.
Line 127: Computational domain: The computational domain is also well presented. I appreciate that the necessary simplifications are systematically mentioned and discussed accordingly!
Line 143: The modelling broiler chicken’s is precise, well referenced and the practical means to derive the fluxes are also well described.
Line 201: Boundary conditions and numerical methods: This section is clear & well documented, however some references (see below) are missing and need to be mentioned.
Lines 227-229: Give the references to your assertions.
Line 244: Grid independence is precisely and well analyzed.
Line 320, validation for air velocity: CFD calculation for air velocity shows generally good agreements for both air speed and flow parameters. However, one does not know if they correspond to the same type of data than the measurements (1, 2, 3 D, or omnidirectional measurements). Also, the founded error must be related to the experimental measurement precision!
Line 347: The layout of Table 2 must be on the same page as for Table 3.
Line 352: The CFD model validation for temperature & humidity is less precise than for air speed but the explanations for the discrepancies are coherent. Also, a discussion including measurement devices errors is missing.
Line 485: Mention that it is the “Computed indoor temperature …”
Lines 494-495: In conclusion, take also into consideration the results of the precision of the experimental measurements!
General comments
This article on the digital modeling of the climate prevailing in a chicken farmis both well presented, well written and its numerical developments are trustworthy
because they are realist as they consider the coupling with the chickens and the
modeling results are validated with respect to experimental measurements carried out
in full-scale commercial shelters. In particular, the coupling of climate with the
physiological activity of chickens is interesting. However, we would have liked it to be
used to determine the physiological comfort zones of chickens based on the external
climate and the adjustments of the ventilation systems. Another contentious point
concerns the absence of description of the systems for measuring both the air speed
but also its temperature and relative humidity together with the precision associated
with these measurements. Also, taking these errors into account in the comparison
between measured and modeled results must be done.
Author Response
Comments 1: Line 102: Experimental farm: The description of the experimental farm is precise and well-illustrated while concise. However, it lacks the description and precision of the microclimate measurements and particularly the types of devices used for air speed determination. Particularly, are they 1, 2, 3 dimensional or omnidirectional (only the modulus of air speed is measured?). Give also the characteristics and the precisions of the air temperature and humidity measurements and use these data in the discussion of the model errors range.
Response 1: Thank you for pointing this out. As the reviewer noted, the explanation regarding the measurements was insufficient, so we have provided additional details. The temperature was measured with a resolution of 0.1°C and an accuracy of ±0.2°C. Humidity measurements were not taken indoors and are therefore excluded. Air speed was measured using a hot-wire sensor (Testo 405i), with a resolution of 0.01 m/s and an accuracy of 0.1 m/s. Since the hot-wire sensor is primarily one-directional, we measured the airspeed in the longitudinal direction to align with the tunnel ventilation flow. Additionally, we have included a discussion on the impact of the anemometer's accuracy during the CFD validation process. We have revised and supplemented the manuscript with this information accordingly.
“Air temperature was measured in the middle of each zone at a height of 0.6 m using thermometer sensors with a resolution of 0.1°C and an accuracy of ±0.2°C”
“While operating five or 14 tunnel fans, the air velocity, in the longitudinal direction to align with the tunnel ventilation flow, was measured at the centre of each of the six zones at a height of 0.6 meters from the floor.”
“Hot-wire anemometers with a resolution of 0.01 m s-1 and an accuracy of 0.1 m s-1 (Testo 405i, Testo Instruments, Germany) were used in all measurements.”
“On average, the predicted indoor air velocities from the CFD simulations demonstrated reasonable accuracy, with errors ranging from approximately -0.2 to 0.3 m s-1. Particularly, considering the anemometer's accuracy of 0.1 m s-1, these results are within an acceptable range.”
Comments 2: Lines 227-229: Give the references to your assertions.
Response 2: Thank you for pointing this out. The parameters related to the performance of the cooling pads, particularly the inertial resistance, had no available references to consult. Therefore, we applied multiple values iteratively and found that 23.3 m⁻¹ provided the best match. As shown in Table 3, this value closely aligned with the measured data. Since this process was based on trial and error, we did not include it explicitly in the paper. The cooling efficiency of 0.7 was assumed, as the typical efficiency of cooling pads is generally considered to be around 70%, which is why we used this value. We have revised the sentence as follows to clarify the meaning.
“In addition, the inertial resistance for the cooling pad performance parameters was deter-mined to be 23.3 m⁻¹ through trial and error by comparing with the measurement results described in Section 2.4. The cooling efficiency was assumed to be 0.7, as cooling pads typically have an efficiency of around 70%.”
Comments 3: Line 320, validation for air velocity: CFD calculation for air velocity shows generally good agreements for both air speed and flow parameters. However, one does not know if they correspond to the same type of data than the measurements (1, 2, 3 D, or omnidirectional measurements). Also, the founded error must be related to the experimental measurement precision!
Response 3: Thank you for pointing this out. We agree with the reviewer's comment that the same type of data should be used when comparing measurements with CFD results. The measurements were conducted using a hot-wire anemometer, which measured air velocity along the length of the facility. Although hot-wire anemometers are generally considered to measure 1-dimensional airflow, they do not strictly measure exact 1-dimensional velocity. Based on our wind tunnel tests, we found that wind speed measurements remained almost consistent when the angle between the wind direction and the measurement axis was within approximately ±45 degrees. Since these results have not been published, we am unable to provide a reference, but we believe this observation may be agreed with professionals who frequently use such anemometers. In the broiler house, the airflow is primarily directed along the length of the facility due to tunnel ventilation, so it can be regarded as measuring the air speed magnitude. Therefore, in the CFD results, we calculated and compared the air speed magnitude with the measured values. Although the sensor's measurement error is relatively small at 0.1 m/s, we have included a discussion in the paper about how this error may influence the comparison with CFD model accuracy, as shown below.
“On average, the predicted indoor air velocities from the CFD simulations demonstrated reasonable accuracy, with errors ranging from approximately -0.2 to 0.3 m s-1. Particularly, considering the anemometer's accuracy of 0.1 m s-1, these results are within an acceptable range.”
Comments 4: Line 347: The layout of Table 2 must be on the same page as for Table 3.
Response 4: Thank you for pointing this out. We have revised the layout of Table 2 to be consistent with Table 3.
Table 2. Comparison of air velocity at the centre of six zones, with 5 and 14 tunnel fans operating (unit: m s-1).
Locations |
5 Tunnel fans |
|
14 Tunnel fans |
|
||
Measured |
CFD result |
Error |
Measured |
CFD result |
Error |
|
Zone 1 |
0.25 |
0.29 |
-0.04 |
1.38 |
1.06 |
0.32 |
Zone 2 |
0.5 |
0.51 |
-0.01 |
1.98 |
1.77 |
0.21 |
Zone 3 |
0.68 |
0.9 |
-0.22 |
2 |
2.04 |
-0.04 |
Zone 4 |
0.33 |
0.28 |
0.05 |
1.33 |
1.01 |
0.32 |
Zone 5 |
0.57 |
0.62 |
-0.05 |
1.82 |
1.77 |
0.04 |
Zone 6 |
0.84 |
0.9 |
-0.06 |
2.11 |
2 |
0.11 |
Comments 5: Line 352: The CFD model validation for temperature & humidity is less precise than for air speed but the explanations for the discrepancies are coherent. Also, a discussion including measurement devices errors is missing.
Response 5: Thank you for pointing this out. The validation of temperature changes was based on transient variations according to ventilation control adjustments, and as a result, the accuracy of the CFD model was slightly lower compared to the case of airspeed. One of the causes of this slightly larger error was that the temperature differences between the six zones were smaller in the CFD model than in the actual measurements. Two main reasons were suggested for this: the possibility of infiltration and the influence of the initial conditions in the CFD model. In our view, the explanation for the discrepancies is adequately explained. As per the reviewer's suggestion, we have also considered the accuracy of the temperature measurement devices when discussing the errors in the CFD model and have revised the paper accordingly.
“In addition, the root-mean-square error (RMSE) between the measured and CFD-computed average temperatures across the six zones was calculated as 0.50 °C. Considering the thermometer’s accuracy of ±0.2°C, these results are within an acceptable range. This also suggests that the CFD model is sufficiently accurate to simulate indoor temperature response to tunnel fan operation.”
Comments 6: Line 485: Mention that it is the “Computed indoor temperature …”
Response 6: The revisions have been made according to the reviewer's comments.
“Figure 12. Comparison of average computed indoor temperature across six zones for different variable inlet configurations.”
Comments 7: Lines 494-495: In conclusion, take also into consideration the results of the precision of the experimental measurements!
Response 7: Thank you for pointing this out. The revisions have been made according to the reviewer's comments.
“In the validation process, which involved comparing airflow velocities and rates in an empty house as well as thermal processes in a house with chickens, the CFD model’s error was within an acceptable range when compared to the measurement device accuracy, confirming the model's accuracy and reliability.”
Comments 8: General comments: This article on the digital modeling of the climate prevailing in a chicken farm is both well presented, well written and its numerical developments are trustworthy because they are realist as they consider the coupling with the chickens and the modeling results are validated with respect to experimental measurements carried out in full-scale commercial shelters. In particular, the coupling of climate with the physiological activity of chickens is interesting. However, we would have liked it to be used to determine the physiological comfort zones of chickens based on the external climate and the adjustments of the ventilation systems. Another contentious point concerns the absence of description of the systems for measuring both the air speed but also its temperature and relative humidity together with the precision associated with these measurements. Also, taking these errors into account in the comparison between measured and modeled results must be done.
Response 8: Thank you for your comments and suggestions. We believe the concerns regarding the explanation of the measurement devices and their accuracy have been fully addressed in the previous responses. Additionally, your suggestion to consider the physiological comfort zone of chickens is a valuable approach, and we will explore this direction in future research.
Reviewer 3 Report
Comments and Suggestions for Authors
This study developed and validated a computational fluid dynamics (CFD) model to predict temporal changes in indoor air temperature in response to variable ventilation operations in a commercial broiler house. This work can be considered to be published after addressing the following comments.
1. As this work developed a CFD model, the full model including the governing equation should be clearly presented.
2. The procedure to solve the model numerically should also be given.
3. The author should consider more advanced CFD tools such as https://github.com/advanCFD/libROUNDSchemes to improve the efficiency of computation.
4. What's R2 in Table 1?
5. Will the error be reduced in Table 2 with a finer mesh?
Author Response
Comments 1: As this work developed a CFD model, the full model including the governing equation should be clearly presented.
Response 1: Thank you for your comments. While the governing equations of the CFD model are important, Reynolds-averaged Navier-Stokes (RANS) is one of the commonly used governing equations in many CFD studies. To maintain conciseness in the paper, we have opted to reference these equations instead. Since this study focuses more on heat generation by the chickens, we have provided a more detailed explanation of that aspect.
“Furthermore, air flows were solved using the Reynolds-averaged Navier-Stokes (RANS) equations with a pressure-based solver[6,25].”
Comments 2: The procedure to solve the model numerically should also be given.
Response 2: Thank you for your comment. We have used a well-established numerical approach to solve the CFD model, which is based on the finite volume method (FVM) for discretizing the governing equations, the Reynolds-averaged Navier-Stokes (RANS) equations. The SIMPLE algorithm was employed for pressure-velocity coupling, and second-order upwind schemes were used for the discretization of momentum, energy, and turbulence equations. Convergence criteria were set to ensure residuals for all flow variables reached acceptable limits. The simulation runs transiently with a time step of 1 s. The brief description of the numerical procedure, including the solution methods and discretization schemes, is already provided in Section 2.2.3. Boundary Conditions and Numerical Models.
Comments 3: The author should consider more advanced CFD tools such as https://github.com/advanCFD/libROUNDSchemes to improve the efficiency of computation.
Response 3: Thank you for the suggestion. We am currently using OpenFOAM in other research, but we will also consider the advanced CFD tool you recommended for future studies. Computational efficiency is always an important concern, and if improvements in efficiency allow us to tackle larger and more complex grid problems, it would certainly help enhance the outcomes of the research.
Comments 4: What's R2 in Table 1?
Response 4: Thank you for your question. In Table 1, R² (R-squared) refers to the coefficient of determination. In Table 1, R² was used to assess how well the CFD simulation results with larger meshes align with those with the smallest mesh.
Comments 5: Will the error be reduced in Table 2 with a finer mesh?
Response 5: Thank you for your question. We often share the same curiosity as the reviewer. Based on our experience and understanding, the answer could be yes or no. Fundamentally, a finer mesh should lead to more accurate results. However, if the source of error lies in the complexity and uncertainty of real-world conditions rather than in the mesh size or numerical schemes, then reducing the mesh size alone may not necessarily bring the CFD results closer to the measured values. Therefore, instead of simply minimizing the mesh size, we conducted a grid independence test to determine an optimal mesh size that ensures the CFD results are not influenced by the mesh resolution, and proceeded with the study based on that size.
Round 2
Reviewer 3 Report
Comments and Suggestions for Authors
Accept in present form